# Ecosystem Services Provided by Seaweeds

João Cotas [ID], Louisa Gomes, Diana Pacheco and Leonel Pereira *[ID]

Department of Life Sciences, MARE-Marine and Environmental Sciences Centre/ARNET—Aquatic Research Network, University of Coimbra, 3000-456 Coimbra, Portugal

* Correspondence: leonel@bot.uc.pt; Tel.: +351-239-855-229

**Abstract:** The ecosystem services can be divided using two major classification systems, the Millennium Ecosystem Assessment (MEA) and the Common International Classification of Ecosystem Services (CICES). In the MEA system, the ecosystem services are divided into four major service clusters: supporting, provisioning, regulating, and cultural. On the other hand, the CICES system regards the "MEA supporting services" as organism natural function (and not an ecosystem service). Thus, this function is the basis for all the three CICES ecosystem services (provisioning, regulating, and cultural) provided by one organism. These ecosystem services can be analyzed for the type of habitat, fauna or flora. Seaweeds, or marine macroalgae, are one of the key organisms in estuarine and seawater habitats ecosystems, which currently is of extreme importance due to the climate changes and the blue–green economy. Seaweeds and humankind have been interlinked from the beginning, mainly as a food source, fibers, biochemicals, natural medicine, ornamental resources, art inspiration, and esthetic values in several coastal communities. Moreover, currently they are being studied as green carbon, carbon sequestration, and as a possible source for the biomedical and pharmaceutical areas. This review is a concise review of all ecological services provided by seaweeds and their impact in the human life and maintenance of the ecosystem status quo. The future of seaweeds use is also approached, regarding the promotion of seaweed ecological services and its dangers in the future.

**Keywords:** seaweeds; ecosystem services; invasive species; ecological functions; seaweed services

## 1. Introduction

Seaweeds or seaweed are macroscopic algae that can grow several meters long (some seaweeds can measure up to 65 m in length). Because seaweed are primary producers, they are at the bottom of the marine food chain [1,2]. Seaweeds are found in aquatic systems with some salinity levels (above 3 PSU) and with sunlight for photosynthesis to take place; these are the two environmental conditions that govern their ecology [1,2]. A solid attachment point is crucial, thus seaweeds prefer to live in the coastal zone (waters near the shore) on stony surfaces, although they can also float. Furthermore, seaweeds play an ecological role in aquatic environments, contributing to oxygen generation, providing a nursery habitat for a variety of marine species, and supplying food for a variety of herbivores [1,2]. They also sustain herbivorous animal communities (invertebrates, such as certain sea urchins and/or gastropods, and vertebrates, such as herbivorous fish) and provide shelter from carnivorous predators. They are a key in the aquatic complex trophic web [3,4].

Seaweeds are the key for the development of submerged vegetation habitats in deep-sea, coastal, and estuarine environments, where the only restriction to the growth and establishment of this communities are light and nutrients [2,5–7]. Thus, it was verified at the Bahamas that there are seaweeds communities at depths of 295 m, although most communities of seaweed are found depths above the 100 m [8]. These environments are considered the most productive habitats on the Earth, providing critical ecosystem for the surviving and subsistence of a large range of animals, including commercial organisms [2,5,6]. Seaweed provide essential coastal protection, uptake of nutrients, and carbon storage [2,5,6]. Furthermore, seaweeds are important for various human activities, as seaweeds are a food

source for several human communities, feed for animals, and fertilizer in agriculture [9–11]. Furthermore, several seaweed compounds are already commercially exploited in human products, such drugs, orthopedics, processed food, and dentistry [7,12,13].

Despite of these above cited seaweed roles and impact on the ecosystem, the seaweed-based ecosystems face major threats by various problems, mostly from anthropogenic factors: human industrial and commercial activities, coastal development, pollution of the aquatic habitats and environments, climate changes, sea level rise, decrease in water quality, and occurrence of invasive of species [2,14–16]. Thus, these non-natural disturbances affect seaweed forests, mostly the canopy-forming seaweeds, commonly known as kelps (large brown seaweeds), promoting appearance of opportunistic and invasive organisms [14,16], consequently affecting ecosystem services that seaweeds provide, generally mostly the coastal protection and as a nursery for commercial species [17]. Due to the development today of seaweed habitats dynamics, there is a need to understand all valuable ecosystem services provided by seaweed to make possible a full analysis of all the value provided by them, especially their economic (food, feed, and compounds raw source) and non-economic values (carbon sequestration, oxygen producer, coastal protection, fish nursery, and water quality) [2,17]. Thus, this review focuses on the wild seaweeds ecosystem services and the problematics from the ecosystem and anthropogenic development. Because they can be a keystone for the sustainable development, we need to understand all the impacts of seaweeds and their implications in a general overview. There are very good and comprehensive revisions on seaweed aquaculture ecosystem services such as those by Kim et al. [18], Hasselström et al. [19], van den Burg et al. [20], Chopin and Tacon [21], and Duarte et al. [22].

Data were collected from online databases, mainly Web of Science, Google Scholar, Science Direct, and Scopus, considering research articles, books, chapters, news, websites, and reviews. The selected topics included the following combinations: seaweed, macroalgae, ecosystem, and ecosystem services: supporting, provisioning, regulating, and cultural. Moreover, other keywords may specific, such as seaweed oxygen production estimation, and seaweed environmental impact. We attempted to obtain the maximum data with scientific support to be analyzed

## 2. A General Overview of Ecosystem Services Provided by Seaweeds

Ecosystem services are "the human gains obtained from ecosystems" according to the Millennium Ecosystem Assessment (MEA). They are divided into four categories: supporting, provisioning, regulating, and cultural services [23,24]. Where Common International Classification of Ecosystem Services (CICES) have the supporting services changed for organism functions, and the services are divided into three categories: provisioning, regulating, and cultural services [25,26]. Although MEA and CICES are two different ecosystem services classification systems, they are similar on their bases, where the MEA is divided in four categories and CICES is divided in three categories.

Supporting services/function are "those required for the development of all other ecosystem services" (Table 1), thus organism natural function due to be basis of all the services and it is all interlinked at the basis. These organism function/services constitute the foundation for all others, and their benefits to people are indirect or occur over time (per example, primary production and photosynthesis). Furthermore, without these organism functions, all the other ecosystem services do not happen [25].

The advantages derived through the management of ecological processes, such as the regulation of climate, water, and some human illnesses are referred to as regulating services. Provisioning services are "products obtained from ecosystems, such as genetic resources, food and fiber, and fresh water," whereas cultural services are "the non-material benefits people obtain from ecosystems, such as cognitive development, reflection, recreation, and esthetic experience" [27].

Although seaweeds demonstrate activity in various ecological services (Tables 1 and 2; Figure 1), some of them are indirect and others affect the local community strictly, thus they

are local ecological services. Few studies have assessed the economic worth of seaweed ecosystem services [28]; however, kelp forests are expected to provide roughly USD 1 million per kilometer of shoreline [16]. If indirect and non-use values are fully examined, the real value is expected to be orders of magnitude larger [28,29].

**Table 1.** Seaweeds functions (CICES)/supporting services (MEA) (according to Reid [23] and CICES [25,26]), and examples of benefits for humans.

| Ecosystem Service Category | Ecosystem Services Provided by Seaweeds | Examples of Benefits for Humans |
|---|---|---|
| Functions (CICES)/ Supporting services (MEA) | Soil formation (sediment formation) | Increase environment quality for aquatic plants, increase in Seaweed primary productivity |
| | Photosynthesis | Production of oxygen and biomass |
| | Primary production | Biomass for higher trophic levels |
| | Production of oxygen | Providing of suitable habitat for fish and other organisms used by humans |
| | Nutrient cycling | Preservation of ecosystem functioning; indirect benefits to food webs and water purification |
| | Water cycling | Influence seawater balance |
| | Provisioning of habitat | Conservation of biodiversity and biomass production of higher trophic levels (e.g., fish) |

**Table 2.** Ecosystem services categories and ecosystem services provided by seaweeds (according to Reid [23] and CICES [25,26]), and examples of benefits for humans.

| Ecosystem Service Category | Ecosystem Services Provided by Seaweeds | Examples of Benefits for Humans |
|---|---|---|
| Regulating services | Climate regulation | Ameliorate the global climatic change |
| | Erosion regulation | Coastal and shore protection and improvement of water quality |
| | Water purification and waste treatment—both in nature as well as in treatment plants | Enhancement of water quality by reducing nutrients and pollutants |
| | Genetic resources | Conveyance of varieties that help aquaculture |
| | Disease | Improvement of water quality by decreasing pathogens |
| | Environmental monitoring | Indication of water quality, pollution and community integrity |
| | Production of atmospheric oxygen | Seaweeds captures the $CO_2$ (dissolved in water) to produce carbon-based molecules and atmospheric $O_2$ |
| Provisioning services | Fiber | Conveyance of a number of types of products |
| | Food | Delivery of sea crops and wild plant products |
| | Biochemicals, natural medicines, and pharmaceuticals | Conveyance of important compounds for human welfare |
| Cultural services | Educational value | Benefits human development and critical thinking |
| | Esthetic values | Aids human introspective development |
| | Recreation and ecotourism | Economic and other benefits for society and for local populations |
| | Cultural heritage values | Aids human introspective development |
| | Inspiration | Aids human introspective development |
| | Local knowledge system | Benefits social welfare |
| | Spiritual and religious services | Nonphysical benefits |

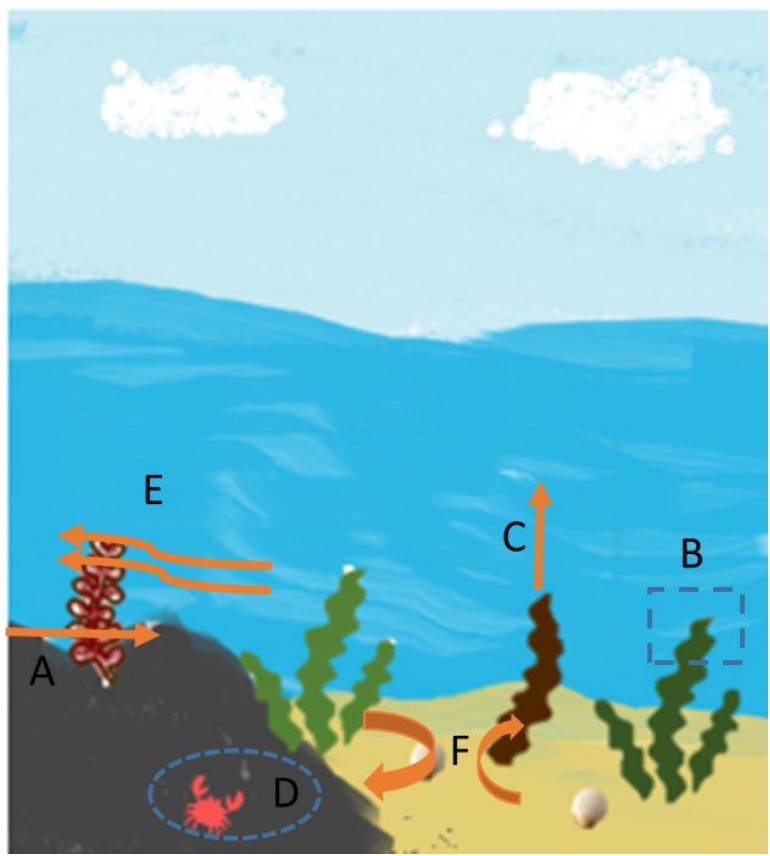

**Figure 1.** Schematic presentation of some of seaweed ecological services: (**A**) sediment formation (supporting services/function); (**B**) biomass production (provisioning services); (**C**) photosynthesis and production of oxygen (regulating services); (**D**) provisioning of habitat (supporting services/function); (**E**) erosion regulation (supporting services/function); (**F**) nutrient cycling (supporting services/function).

Although seaweeds have a wide range of ecosystem services, there is scarce literature about the economic values of these ecosystem services, where the kelp forests are the most researched seaweed community. In addition, there are scientific reports measuring their economic impact as demonstrated in Table 3. Most of the economic importance is on provisioning services, such as human food and also in biochemical, natural medicines, and pharmaceutical areas [9]. Thus, seaweed cultivation is gaining interest currently, evidenced by the studies of Kim et al. [18], Hasselström et al. [19], van den Burg et al. [20], Chopin and Tacon [21], and Duarte et al. [22], reviews and studies on seaweed cultivation economic and ecosystem impacts. On the other hand, there is general lack of value placed on wild seaweed and its economic impact and its impact on the ecosystem.

**Table 3.** Economic analysis of kelp forests ecosystem services.

| Region | Ecosystem Service | Economic Value (US Dollars per Kilometer per Year) | Reference |
|---|---|---|---|
| Pacific | Provisioning services: 91%; Others ecosystem services: 9% | 811,000 | Vásquez et al. [30] |
| Great Southern Reef | Cultural services: 90 %; Provisioning services: 10% | 914,000 | Bennett et al. [28] |
| South Atlantic Ocean | Provisioning services: 45%; Cultural services: 30 %; Others ecosystem services: 25% | 520,000 | Blamey and Bolton [31] |

## 3. Seaweed Ecosystem Services

The ecological services importance of seaweeds is mainly linked to the fact that these organisms are inherently potent bioaccumulators of heavy metals, noxious chemicals (such as dibenzodioxins and other poisons), and even microplastics [32,33]. Seaweeds contribute to systematic marine bioremediation and carbon sequestration, as well as reducing ocean acidification [34]. Furthermore, seaweeds sustain complex food webs in marine ecosystems and lessen wave pressures on beaches, lowering the danger of coastal disasters. Furthermore, seaweeds can even mitigate algal blooms by regulating nitrate and phosphate balances [35]. All these features combine to make seaweed a multivalent and highly adaptable organism to its surroundings [9].

Similarly, macroalgal forests provide a variety of ecological services, including direct support for commercial, recreational, and subsistence fisheries and aquaculture [28,30,31,36]. Erosion management and temperature change are examples of indirect ecological services. Cultural and religious importance, biodiversity, and scientific worth are all examples of intrinsic ecological services. Few studies have assessed the economic worth of macroalgal ecosystem services [28]; however, kelp forests are expected to provide roughly USD 1 million per km of shoreline [16]. If indirect and non-use values are fully examined, the real value is likely to be orders of magnitude larger [28,29].

### 3.1. Seaweed Functions/Supporting Services

There is significant evidence that seaweeds play critical roles in supplying all categories of supporting services (Table 1). Sediment formation is influenced by seaweeds, mostly seaweed forests. Sediments are formed of particles of both allochthonous and autochthonous origin, with seaweed detritus playing a prominent role in the latter. Seaweeds are the principal source of biological materials in coastal areas. Kelps (large brown seaweeds) limit water velocity, which aids in the trapping of mud and sand in beaches. For these reasons, seaweeds aid in sediment retention and deposition [37–40]. The correlation between wind and storms indicates that climate change will most likely has a significant influence on wave height and other wave characteristics [41]. Floods and coastal erosion are severe threats to coastal communities, with occurrences that require the implementation of long-term solutions to the problem [42]. By providing coastal protection, vegetated coastal habitats may assist to alleviate the effects of sea-level rise and the concomitant increase in wave activity [43]. Furthermore, vegetated coastal habitats have a high capacity for producing carbonates and other materials that aid in sediment accumulation, beach nutrition, and the formation of dunes on land [44,45], preventing coastal erosion.

Photosynthesis and primary production by seaweeds provide wide indirect advantages to humans. Photosynthesis and primary production of seaweeds, for example, feed aquatic food webs in a variety of environments. Seaweeds environments are very productive, producing 1.5 and 1.05 g C m$^{-2}$ h$^{-1}$ (median values) by photosynthesis, respectively [46]. On a worldwide scale, seaweeds habitats (such as kelp forests and seaweed beds) are the most widespread and productive coastal ecosystems, and are one of the principal photosynthesis organisms in planet Earth [47–49]. Much of this carbon is used to support the food web in coastal habitats, with grazers consuming on average 34% of seaweed production [46]. Seaweeds, on the other hand, are more nutritious (compared with other seaweeds) and hence vulnerable to more grazing, particularly in temperate climates, while their tropical counterparts tend to contain more anti-herbivory chemicals [50]. The detrital cycle absorbs around 38% of the carbon generated, as seen in [46]. Once the residual seaweed biomass enters the detrital cycle, bacteria and fauna break down or re-mineralize seaweed carbon into $CO_2$, which is then converted or absorbed into new biomass. Seaweeds, in addition to providing organic matter for detritivorous species, also offer food for aquatic and terrestrial herbivores [51,52]. Thus, seaweeds play an essential role in the food webs of saltwater ecosystems through photosynthesis and primary production, which translates into a rise in the production of fish, abalone, sea urchins, and other animals

consumed by humans (provisioning services). Seaweeds are the feed in the growth of herbivores and permit the food chain maintain its status quo.

Photosynthesis also contributes to another important supporting service: production of atmospheric oxygen. Seaweeds belong to the aquatic photosynthetic group (this group is composed of seaweeds, marine plants, and phytoplankton) responsible for 50–80% of the oxygen production on Earth. In comparison, only 28% of the oxygen production on Earth is from rainforests. However, the numbers are continually fluctuating, and calculating the precise proportion of oxygen generated in the water is challenging [53]. By comparing the quantity of organic carbon created per square meter on an annual basis, we may compare seaweeds oxygen production to that of terrestrial plants. Seaweeds can produce 2 to 14 kg of biomass, but terrestrial plants, such as trees and grasses in temperate regions, and microalgae can only produce approximately 1 kg [54]. Furthermore, seaweeds are estimated capture around 175 million tons of carbon, accounting for 10% of global automobile emissions [55].

Oxygen is required for decomposing microbes to metabolize toxins and organic pollutants [56,57], and it is a critical component in nitrogen, phosphorus, sulfur, and iron cycling [58]. Thus, seaweed photosynthesis helps in effective nutrient cycling. Production of oxygen by seaweed is also vital for fish, particularly those more sensitive species [59], and so seaweeds may have an indirect impact on human usage of this resource.

The influence on nutrient cycling and on aquatic life survival are only two of the many indirect advantages supplied by oxygen generation. Seaweeds have a significant impact on nutrient cycling through a variety of physical, chemical, and metabolic processes, as well as interactions with other species [38,60,61]. Kelps (large brown seaweeds), for example act as physical barriers against sea currents (and wave disturbance), reducing sediment re-suspension, and also reduce water velocity, increasing particle and phosphorus sedimentation [38,60–62]. These physical factors contribute to the reduction in nutrients in the water column. Seaweeds, together with sediment, are the primary component in which nitrogen and phosphorus are stored in a variety of water bodies [62,63], As a result, alterations in this population have an impact on the metabolism of the entire aquatic environment. Seaweeds also have an impact on nutrient cycling because they absorb nutrients and after death, they liberate the nutrient to the water column through biomass decomposition. Thus, by interfering with nutrient cycling, seaweeds have a direct impact on phytoplankton and periphyton primary production, as well as water quality [64].

Seaweeds participate in the water cycle as a result of two mechanisms. Seaweeds can be exposed by different tides and if there is a low tide, they can be emerged from the aquatic habitat and being exposed to air, provoking water losses. The opposite occurs when seaweed are submerged, when they reduce water losses to the environment. The magnitude of these processes interferes with the water cycle and may have a crucial influence in water balance in specific ecosystems.

The provisioning of habitat can be considered one of the most important ESs provided by seaweeds because a variety of other ecological services depend on this service. Seaweeds are vital for maintaining the food chain's equilibrium and for ecological functions such as nursery, recruitment, and protection regions for many aquatic species [19,65]. Natural macroalgal niches perform an important ecological function in marine environments [19,65] (Figure 2), and as a result, are a significant component in economic exploration and environmental conservation of the oceans and seas. As a result, preserving indigenous seaweed in their natural environments is critical, as is avoiding the introduction of foreign species, which can jeopardize ecological stability. However, to facilitate the use of marine resources, eco-sustainable seaweed production may be used to ensure that this exploitation does not impose strain on marine environments. The growth of native seaweed techniques has the potential to provide immediate assistance for the many fish, crustaceans, cephalopods, and echinoderm species that have been discovered [19,65]. Thus, seaweeds provide aquatic animal habitats which are explored by the fisheries (for human food, mainly, this also is well-demonstrated by offshore seaweed farm in which fish catch is growing). The example

of seaweed traditional farming in Indonesia is a perfect example of this seaweed ecosystem service benefit [9,66].

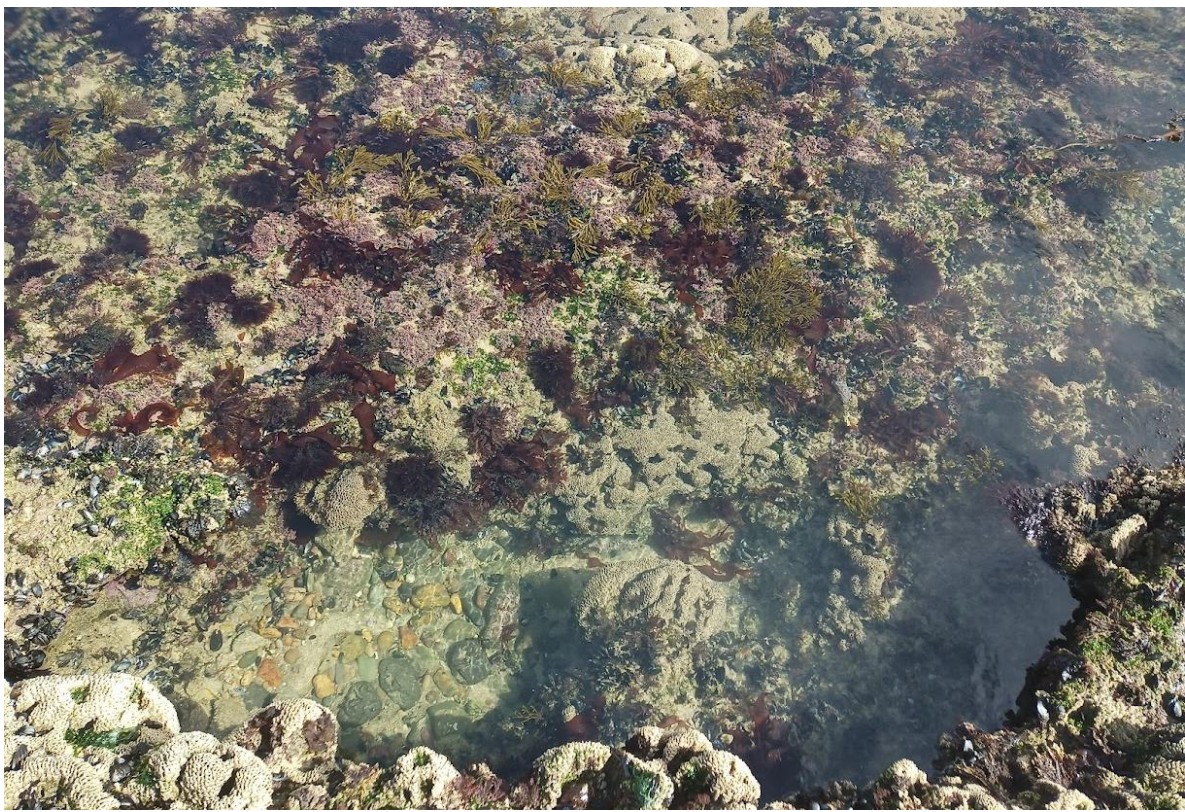

**Figure 2.** Ecological niche with seaweeds and other marine organisms.

*3.2. Regulating Services*

Unlike supporting services, regulating services provide advantages that are more directly tied to human benefits; however, they are not always clearly recognized such as the climate regulation provided by seaweeds. The climate problem worsens while the international community is attempting to make the commitment required to reverse it, for example the promotion of the Sustainable Development Goals by ONU [2]. The climate problem worsens while the international community refuses to make the entire commitment required to reverse it. The year 2019 was the second warmest year on record, as well as the conclusion of the warmest decade (2010–2019), resulting in catastrophic wildfires, storms, droughts, floods, and other weather disasters all over the world. Temperatures throughout the world are expected to climb by 3.2 °C by the end of the century. To attain the Paris Agreement's highest objective of 1.5 °C, or even 2 °C, greenhouse gas emissions must begin to decline by 7.6% per year beginning in 2020 [2].

Highly prolific seaweed species may contribute significantly to the yearly biological reduction in $CO_2$ and the global carbon cycle [67]. However, in order to comprehend the scope of this decrease, researchers must first estimate the amount and rate at which fixed carbon is recycled [68]. Even though marine seaweed communities comprise a very tiny portion of the coastal region (approximately 33%, 3,5 mill km$^2$), these habitats are an important component of climatic change adaptation and mitigation strategy [69]. Thus, seaweed are responsible for the uptake of 173 million tons of carbon per year, making seaweed a vital player in the global carbon cycle [48]. Seaweed are carried to the deep ocean by drifting and sinking after being pulled away from the substrate [48]. Fresh *Sargassum* were discovered in the intestines of abyssal isopods (a kind of marine crustacean) residing on the bottom at 6.475 m depth, indicating the transfer of seaweed to the deep

waters. Organic material is considered gone from the carbon cycle when it falls below 1.000 m [48]. Seaweed also mitigates acidification, deoxygenation, and other marine effects of global warming, which endanger sea biodiversity and the source of food and livelihood for hundreds of millions of people [70]. Thus, the promotion of sustainable seaweed cultivation can promote this service even more. Seaweed cultivation can assist to prevent ocean acidification and deoxygenation [71]. Deoxygenation has arisen as a concern for ocean warming and its impact on ecosystems, posing a challenge to the mitigation strategy for climate change [72]. Hypoxic conditions are especially dangerous in eutrophic marine settings, endangering the life of local flora and fauna. These locations can profit from seaweed farming since they are autotrophic ecosystems that produce oxygen through photosynthesis. Furthermore, seaweed farming has the ability to improve the pH balance of ocean water and provide oxygen, assisting marine life in adapting to warmer waters [71].

Erosion regulation is also linked to seaweed colonization. Seaweeds prevent erosion and sediment trapping, aiding in the maintenance of saltwater environments' shorelines (Figure 3). Shore erosion is reduced because seaweeds act as physical barriers that minimize wave energy [39,73].This is also observed in estuarine seaweeds that trap sediment in the river margins [39]. Beach-cast seaweeds may provide indirect coastal protection by releasing nutrients to dune ecosystems, which in turn stabilize local sediments and contribute to coastal protection [74]. There is little scientific evidence to imply that drift deposits on beaches actually provide coastal protection. However, if present on a big enough scale, they may be able to diffuse wave energy, shielding sediments from wave scour on a small scale [39].

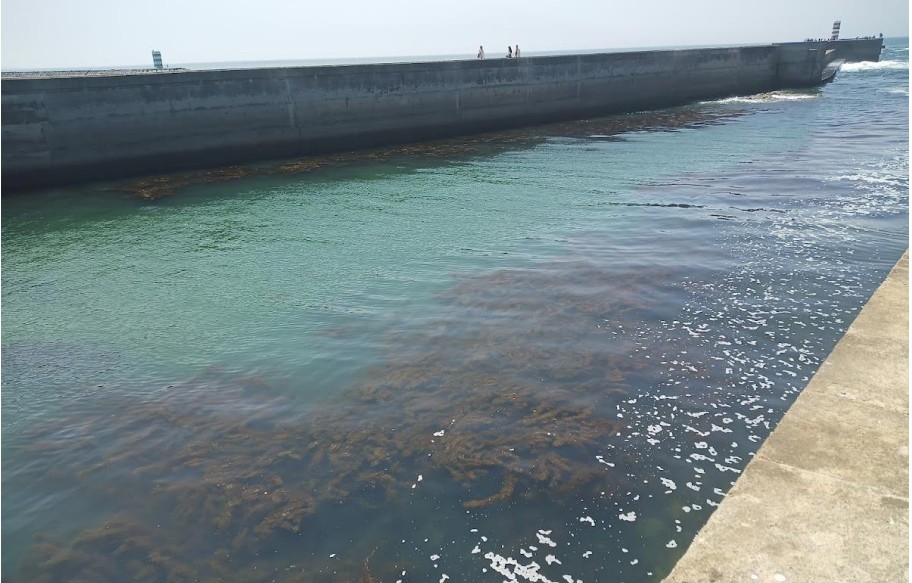

**Figure 3.** Seaweed beds on aquatic habitat reducing the wave effect in coastal shore, trapping the sedimentation and reducing coastal erosion.

Access to drinkable and high-quality water is critical for protecting human health and ecosystems. Seaweeds communities are already phytoremediators in their native settings, thus being water purification systems [10]. As a result, seaweeds may be used to treat polluted waters or wastewaters from various sources with varying salinities, such as estuarine and marine settings [75]. As a result, seaweeds can play an important role in improving water quality and reducing pollution. In sectors such as aquaculture, seaweeds are already employed to boost water efficiency. Seaweeds, on the other hand, serve as a biofilter in the ecosystem. Metal-contaminated aquatic environments are becoming a life-threatening hazard in various locations as a result of rapid urbanization and economic expansion (anthropogenic activities). As a result, transportation can raise that pressure even higher, to the point where the environment suffers the most harm as a result of this

contamination [75]. During their development period, seaweed may deposit or bio-adsorb metals in their cell walls and sequester these toxic compounds from the surrounding environment, detoxifying the aquatic system and allowing the ecosystem to retain its usual status quo [75].

Because of their high metal accumulation capabilities, wild seaweeds are regarded as natural metals biosorbents. For example, brown seaweed are known to take Lead, Cadmium, Copper, Zinc, Chromium, and Cobalt from industrial effluents or wastewaters [76]. However, metal absorption in seaweed is generally accomplished by the interaction of metal ions with algal cell walls; therefore, metal removal is not metabolically regulated [76].

Changes in ecosystems can alter pathogen abundance, which is connected to disease regulation. Seaweeds (together with its associated microbiome) act as a filter for allochthonous materials, which helps to reduce infections. Pathogen reduction strategies include adsorption with other particles, adsorption, predation by bacteria in the biofilm, and exposure to seaweeds and other microorganism exudates, such phenolic compounds and sulphated polysaccharides [50,77,78].

*3.3. Provisioning Services*

The most immediately recognized services are the direct usage of seaweeds products as food, textiles, medications, and decorations (provisioning services). These provisioning services are provided by four distinct sources. Seaweed aquaculture in the supply chain is used more in the provisioning service (Figure 4). The second supply source is related with seaweeds taken directly from nature, the biomass of which is used for a variety of uses by local populations [9]. A third source is the biomass of seaweed, which grows abundantly in nature, and in this instance, harvesting and using their biomass represents a solution to a problem (which occurs with invasive species, for example) [14].

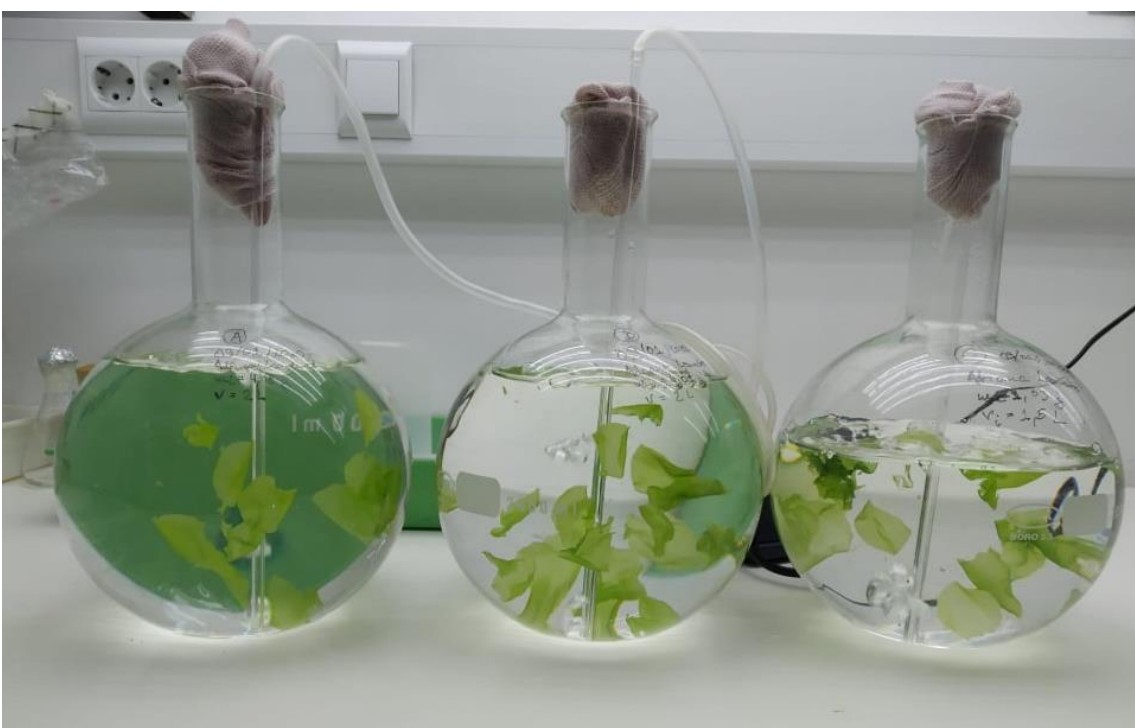

**Figure 4.** Seaweed juvenile cultivation for aquaculture upscaling.

Edible seaweeds have been consumed by coastal cultures all over the world since prehistoric times. Currently, seaweed are a common food source in the diet in many Asian nations, and their popularity is expanding in Western cultures, owing to both the inflow of Asian cuisine and the well-known health advantages connected with their use [79].

Seaweeds are mostly grown for human use, with over 20 thousand tons produced in 2016, primarily *Saccharina japonica* (formerly *Laminaria japonica*), *Undaria pinnatifida* (brown seaweeds), and *Porphyra* sp. (red seaweed). They are mostly consumed in Japan, China, and Korea [9].

Seaweed are consumed as a food supplement due to their nutritional profile, which is rich in dietary fibers, proteins, and minerals necessary for the proper functioning of human cell mechanisms [80]; however, due to mineral limitations, the recommended daily intake is only 5 g of dried biomass per day [81]. Epidemiological data show that consuming seaweed on a regular basis may protect against a range of contemporary illnesses such as cancer incidence and cardiovascular disease [80,81]. In acute feeding experiments in humans, the addition of seaweed to diets previously demonstrated the ability to promote satiety and lower postprandial glucose and fat absorption rates [79]. Regular eating of seaweed can provide a nutrient-dense boost to the diet (carbohydrates, proteins, and vitamins). However, nutritionists may be concerned if micronutrient consumption exceeds the Reference Nutrient Intake (RNI), especially if bioavailability is high [79]. Beyond its basic macronutrient content, seaweed is now a significant industry research and development idea as a nutraceutical or functional food with dietary benefits [7].

Since ancient times, seaweed has been used as animal food in coastal areas. Seaweeds are blended with animal feed since they might be harmful to animals if taken alone. Seaweed biomass is an important alternative feed element for cattle. The nutritional value of seaweed, combined with their non-animal nature, makes them particularly suitable for use in animal feed as nutraceuticals, a term derived from the combination of nutritional and pharmaceutical, used to identify food components that provide health benefits including disease prevention [82,83].

In vitro studies and certain animal studies have supported the health advantages of seaweed beyond the supply of vital nutrients; however, many of these studies have used inadequate biomarkers to validate a claim and did not move to appropriately design trials to test effectiveness. Some seaweed components are appealing as functional food ingredients based on the limited evidence available, but more animal nutritional studies evidence (including mechanistic evidence) is required to evaluate both the nutritional benefit conferred and the efficacy of purported bioactivities, as well as any potential adverse effects [11,84,85].

Seaweeds are on the verge of becoming popular due to their appropriateness as possible feedstock production as well as dietary supplements. Seaweeds can meet the growing demand for renewable and sustainable energy sources without compromising food and land resources because they are fast growing, high biomass yielding, and elevated and free of charge productivity compared with other conventional biomass feedstock, such as corn or soybean. However, seaweeds demonstrate the potential to be further investigated as an animal feed additive/supplement and cannot be used as a complete replacement for traditional animal feed. Seaweed's beneficial benefits are normally limited to less than 10% of the overall content in animal feed; above that, it has been shown to have detrimental effects, and animals have refused to consume the offered feed [11].

The use of fibers (or biomass) of seaweed also has several applications by humans. The seaweed biomass produced can be exploited for different exploitations, such as fertilizers [86], soil conditioner [87], plastics [88], bricks [89], houses, metal accumulator (dried biomass to perform bioremediation) [10], and antiflame materials (used in textile and others human-related consumables) [12]. Another potential use of seaweed fibers is the production of biofuel. This can be accomplished, for example, by employing seaweed biomass as a biofuel that directly replaces fossil fuels [90,91]. Furthermore, "blue" biofuels derived from seaweed do not compete for resources with agriculture because they do not require arable land, freshwater, or chemical applications (fertilizer, herbicide, or pesticide), and are thus more environmentally friendly in many ways than current biofuels derived from land crops [42].

Genetic resources are an new ecological service produced by seaweeds. Because of the rapid growth of the worldwide seaweed aquaculture business, there is a growing interest in translocating seedlings developed from wild-type brood stock. Such translocations, however, must be used with caution since imported cultivars might impair the genetic structure and variety of wild populations. To inform decision-making about aquaculture translocation initiatives, an understanding of the genetic structure and connectivity of target species is essential [92]. As a result, scientific study on seaweed development is very timely: the capacity of the sector to produce seaweeds with changed morphological traits (e.g., thicker blades), greater growth rates, or delayed (or even no) fertility determines the sector's potential for expansion [93]. Because the architecture of seaweeds (or marine seaweed) is significantly simpler than that of land plants, and because seaweeds belong to three separate eukaryotic groups, the processes governing their morphogenesis are critical to understanding their development [93].

In general, the development of seaweed genetic resources, strain selection, and selective breeding software is in its early stages. Focused in the morphological variations in commercially significant seaweeds and investigate the genetic basis for these morphological alterations [93].

Another major provisioning function offered by seaweeds is the extraction of biochemicals, natural medicines, and pharmaceuticals. In folk medicine, seaweeds have been used for generations as a standard medication for a variety of health problems [94]. This attribute was considered as early as 300 BC in Asian cultures, and Celtic, British, and Roman populations living near the sea used them for 1000 years for healing wounds, as a vermifuge, or as an anthelmintic, so modern research in pharmaceutical and biomedical areas is evolving and progressing [95,96]. Many studies have shown that seaweeds have nutraceutical, pharmacological, and cosmeceutical benefit. They have anti-cancer, antiviral, antifungal, antidiabetic, antihypertensive, immuno-modulatory, cytotoxic antibiotic, anticoagulant, anti-inflammatory, anti-parasitic, antioxidant, UV-protective, and neuroprotective effects [96–101]. It has also been established that several species of seaweed contain powerful antioxidant compounds such as phlorotannins, carotenoids, and sterols, making seaweed a source of compounds with potential neuroprotective effects, which can be useful in the treatment of neurodegenerative diseases such as Parkinson's and Alzheimer's [102,103]. Sulfated polysaccharides from seaweed demonstrated significant potential pharmaceutical applications, such as anti-ulcer actions by reducing the adherence of Helicobacter pylori infection [104]. Actually, various seaweeds compounds are used in various industries, and also in pharmaceutical and biomedical applications to treat human health problems and diseases [12,13,81,105,106].

The ornamental resources by seaweeds are based mostly in aquariums and aquatic gardens and parks [107,108]. However, this type of service can be harmful due to the lack of biosecurity, and the possibility of non-native species [107].

Finally, seaweed have been widely employed in biomonitoring. Although the MEA does not consider biomonitoring to be an Ecological Service by Millennium Ecological Assessment, biomonitoring methods based on seaweeds are being included in the European Water Directive Framework. Furthermore, the presence or absence of seaweed in coastal regions is a helpful tool for evaluating water quality, ecological status and is already being utilized in several nations in this capacity [109–111]. In Portugal, for example, the PMarMAT tool is used to analyze the water quality along rocky coasts under the European Water Framework Directive, and it allows for an examination of algal populations to characterize the water in coastal locations [109].

### 3.4. Cultural Services

Seaweeds are also present in various cultural services. They are present in various traditions, per example, Asian countries, the Celts, and the Romans used seaweed as food, feed, and fertilizer [87,112]. Mattress fillings, roof construction, wall insulation, knife handles, musical instruments, contraceptive devices, infant teethers, hygrometers,

firecrackers, fishing lines, baskets, dolls, jugs, jewelry, and buttons are some more unusual traditional seaweed applications [113]. As a result, it is demonstrated as inevitable that seaweeds and humans have had a plethora of intertwined connections on evolutionary timelines as well as in recent millennia and centuries all the way into the Anthropocene. It is no surprise that seaweeds have invaded and acted as a savior for humanity all over the world in various times of dire need and crisis. Indeed, they have been used as a last option in times of hunger, wars, disease outbreaks, nuclear catastrophes, or as components of preserving the fabric of societal stability. In certain regions of the world, the tradition of utilizing seaweeds as a valued food has survived to the current day [114–116].

Whilst in others, this use was seemingly forgotten, and seaweeds have mostly served as various raw materials for a variety of practical uses and extracts, often in remote ways where most people would not know that a specific application was actually based on a specific type of seaweed, such as a food ingredient, a hydrogel, a medicinal aid, or a fire repellent [112]. Demonstrating that in various human communities seaweed has had high cultural heritage value (Figure 5) since ancient times, there is a seaweed-related cultural heritage notorious in Shandong province (China). In this province there are the so-called "seaweed bungalows" (seaweed are used to cover the rooftop) which is dated from the Qin Dynasty (221–206 BC) and continues today and it is a listed provincial intangible cultural heritage in Shandong [112,117]. Other cultural heritage is gastronomy related, for example in Azores Islands (Portugal) and in Ireland there are recipes with seaweeds (mostly with *Porphyra* sp.) which have been maintained for centuries [118].

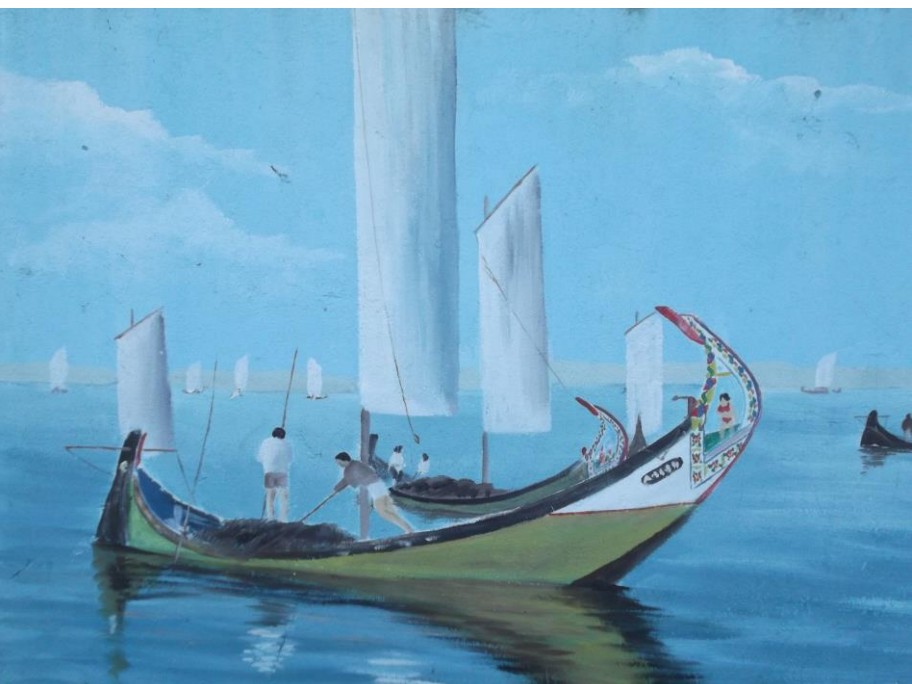

**Figure 5.** House painting representing the "moliço" harvesting at Ria de Aveiro (Murtosa, Aveiro, Portugal), which "moliço" consist in the harvest of seaweeds and aquatic plants to fertilize the agricultural soil, seaweed tradition, and regional cultural heritage.

There are religions which have spiritual and religious values related with ecosystems or their components [23] and spiritual and religious services are also provided by seaweeds. They were highly regarded, even venerated, in the Eastern culture, and were included in daily dinners and were a method of paying taxes to the Japanese Emperor's court or as a gift for guests [54]. Sze Teu, a Chinese scholar, stated around 600 BCE that "certain seaweeds are a delicacy for the most honorable visitors, even the king himself." Seaweeds were so revered that they were one of the Twelve Symbols of Sovereignty, which were painted or embroidered on the official robes used by the Chinese emperors, symbolizing

brightness and purity [115]. In contrast, seaweeds did not have the same exalted position in the majority of the Western world (i.e., Europe and North America) [113]. The Greek deity Nereus, father of the Nereids (beautiful sea-nymphs), was a wise old man who lived in a magnificent grotto in the depths of the Aegean Sea. He had the power of divination and could transform himself into any form. He is seen with seaweed hair and a scepter or trident [119]. Poseidon's son, Triton, had a fishtail coated in fine scales and green seaweed all over his body. He was a merman who ruled the seas, blowing his shell trumpet to terrify giants during battles or to raise or reduce the waves [119]. Thalassa, the primeval Greek goddess of the sea and the genesis of fishes and other sea animals, is shown in Greco-Roman mosaics as a matron half-submerged in the sea, wearing seaweeds, wearing crab-claw horns, and holding a boat's oar [113]. There were also gods/goddesses and mythology associated with various seaweeds in ancient Celtic culture. In contrast to the Greco-Latin culture, most of these tales contained a wide range of animals (both good and bad), and those traditions were utilized to terrify people, particularly youngsters. Because of its spectacular beauty, *Nereocystis luetkeana* (bullwhip kelp) (brown seaweed) plays a vital part in the Northwest Coast people's traditions and folklore. In Haida tradition, a double-headed bullwhip kelp signaled the entrance to the undersea home of a supernatural chief, and anybody who could follow it down to the seabed and meet the supernatural people there was bound to gain power, respect, and good luck when they returned to their hometown [120].

This history of seaweed and humankind demonstrates that seaweed provided a rich source of inspiration for art, folklore, national symbols, and architecture (not only houses but other equipment). This is demonstrated by the strong ties between seaweeds and humans as may be seen in culture, mythology, folklore, and poetry [113,115]. In the current context, humankind and human societies have repeatedly turned to seaweeds in times of crisis to take advantage of what this diverse and ancient, polyphyletic assemblage of marine, photosynthetic organisms can offer in order to meet basic sustenance needs, alleviate disease suffering, and secure health, well-being, and survival at critical stages of human history. With the terrible extent of global challenges that people and civilizations are currently confronted with, such as pandemics, climate change, and the need for sustainable food sources and supplies, we may turn our collective and concentrated attention to seaweeds once more. Seaweeds, with their numerous extraordinary and unique material characteristics and chemical components, have unsurprisingly come to the aid of soldiers throughout times of conflict. An example from history explains how Muslim scientists and naval commanders in the early eight century discovered a means to utilize seaweed to protect their ships from the catastrophic so-called Greek Fire used by the Byzantine navy to set enemy boats on fire [112]. Abd al-Rahmn, an Alexandrian, discovered how to extract "algin" from the brown alga *Gongolaria barbata* (formerly *Cystoseira barbata*) (brown seaweed), which he subsequently used to fireproof fabric. He used this material to shelter the ships from Greek fire [121].

Esthetic values of seaweeds explain why certain civilizations encourage seaweed farming, seaweed-based buildings, "scenic dives in kelp forests," and why many individuals choose residence near these coastal areas or in house surrounded by seaweeds [9,122–124]. Per example, is the Shandong cultural heritage, where seaweeds are used in esthetic on the houses. Furthermore, this happens in others communities, for example in Læsø (Norway), where seaweed are used in house esthetic value [125]. The esthetic values are related to other ecosystem services, recreation and ecotourism, which evolves into an eco-friendly and sustainable mindset. Thus, there is ecotourism developing around seaweeds, for example, restaurants that provide local, seasonal, and sustainable food to entrepreneurs that offer "pick your own seaweed" excursions [126]. There are companies that organize a variety of activities that use seaweeds, such as seaweed paddling. Participants on a kayak day excursion may snorkel and pick natural seaweed in a sustainable manner, as well as learn how to prepare it. How may seaweed become a significant element in the development of a resort through marine experiences? Seaweed and kelp forests are vital to local econ-

omy and tourist activities including animal viewing, fishing, kayaking, snorkeling, and scuba diving. Divers, for example, enjoy the beautiful underwater beauty provided by kelp forests around the beaches and their richness [127,128]. Furthermore, marine eco-tourism is a relatively new and increasing phenomena, with the environmental educational component as well as the minimal environmental effect of these recreational activities at the forefront [129]. Furthermore, in Indonesia, the Philippines, and Malaysia there is the development of seaweed traditional aquaculture service based on tourist recreation of the ancient cultivation techniques [124,130–132]. One of the seaweed cultivation tourism cases, in Olango islands (Philippines), is capable benefit near 10,000 families in direct income. Moreover, it also affects indirect income due to seaweeds being the key for the already exploit tourism objectives, such as diving, snorkeling, and watching migratory birds [131].

There also educational values associated with seaweed. Students may have their first encounter with seaweeds during formal schooling during biology lectures, when they go on field trips to the beaches. Beach field tours on rocky coasts with seaweeds and other aquatic organisms are also used to educate children on the fundamentals of ecology and food webs. Seaweeds are easily seen and hence contribute to the transfer of important ecological ideas and principles. Currently, there is an increase in education-related scientific projects (in Europe Union, specifically) which transmits educational values about the sea ecosystem importance, thus seaweed importance in ecological values. For example, the "Roteiro-entre-Marés" (Figure 6) try to enhance the natural and cultural resources linked to the Portuguese coast, specifically the intertidal zone (inclusively seaweed), using as an example two distinct coastal ecosystems [133]. However, this type of educational value are being promoted by interactive learning and using information technologies to promote and spread ecosystem services for a broader target to promote ecological sense and knowledge [134]. In the USA, there are associations also promoting seaweed educational value, increasing their importance for the humankind, and transmitting ecological values [135].

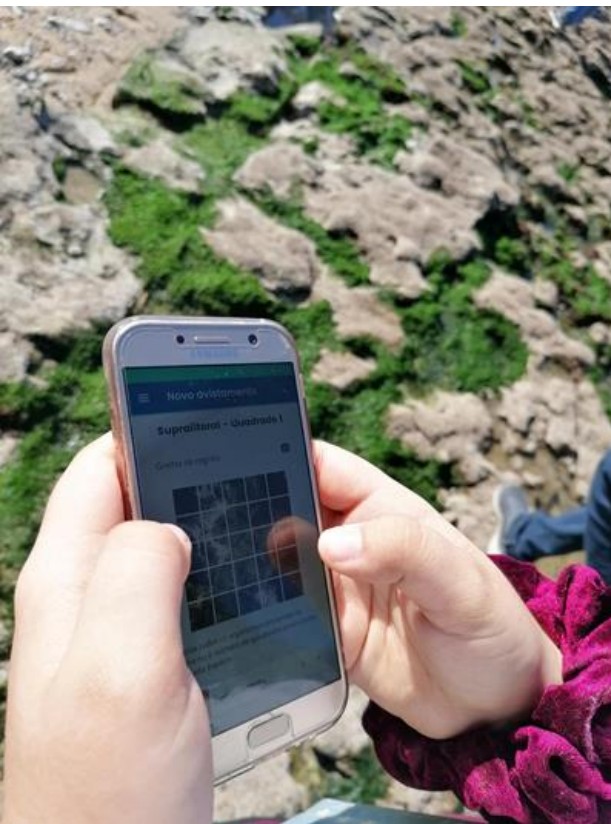

**Figure 6.** Educational program "Roteiro-entre-marés" app regarding seaweeds natural occurrence.

Humans inhabiting places near ecosystems where seaweed dominate, such as coastal areas, may develop local knowledge systems. Central Europe is the second example of a local knowledge system. According to historical records, certain seaweeds in that region have been exploited for grazing by cattle and pigs, trampling, and uprooting for around 300 years [136]. The combination of approaches used in all these activities can be regarded as a component of the local population's knowledge system. This traditional knowledge related with local activities, according to Biro et al. (2019), aids in the creation of creative strategies for environmental protection. Similarly, to this local knowledge system, for a long time on the European Atlantic Coast, seaweeds have been utilized to improve barren agricultural soils or soils along the coastline. The Romans, or the Celtic tribes of Gaul, Brittany, and the British Isles left the oldest written documents in Europe where the information was gathered in a local knowledge system, which is well-documented until the twentieth century. However, this knowledge is being lost due to the death of elder villagers without passing the information to young villagers [87]. However, there is an attempt by the educational values to retrieve the local knowledge system about seaweeds and their ancient/old uses and applications, sometimes using folklores or performing scientific methods to rediscover ancient knowledge, mostly in agriculture and animal feed [11,87,137,138].

Thus, this demonstrates that seaweed have been an inspiration since ancient times, and with a change in mindset, seaweeds can be further developed into the cultural valorization around the globe, although there are some dangers involved. Moreover, it may be difficult to the progress and achievements accomplished in the seaweed world.

## 4. Future of Seaweed Ecological Services

However, these ecological services are in jeopardy owing to the advent of non-native species with invasive behavior in new environments. As a result of the invading species' complete dystrophy, the general ecosystem will be altered to a perilous state. The underlying issue in this situation is not seaweeds species, but anthropogenic actions and activities associated with climate change and eutrophication of aquatic ecosystems. Changes in seaweed distribution, abundance, and community structure caused by climate change will have an impact on ecosystem function. The extinction of canopy-forming species is expected to diminish the availability of a biogenic habitat and the biodiversity it supports, with consequences for food web structure and function [139]. Changes in seaweed standing stocks or biomass, as well as changes in species composition, have an impact on biogeochemical cycling, including inorganic carbon and nitrogen uptake, organic carbon generation, and oxygen evolution [1]. Many of these tasks are difficult to replace, especially when functional redundancy within seaweed communities reduces as seaweed biodiversity declines [140].

These ecological dangers and impacts create a slew of ecological and economic quandaries for the afflicted countries. This is happening in a number of Caribbean nations with *Sargassum muticum*, together with other species of this genus (brown seaweeds) [141]. To offset this detrimental impact, ways to profit from invading macroalgal species must be developed. Though biogas and fertilizer production are examples of ecological services that might benefit from the lowering of the biomass of ecologically harmful species and ensure a sustainable future in these nations [141].

Furthermore, for global food and feed, seaweed production can be a low-cost alternative to minimize ecological pressure [142]. Because of the minimal capital investment, short crop/harvest cycle, and ease of cultivation, seaweed can be utilized as an alternative source of sustenance by coastal human groups in coastal nations, causing a multifunctional primary product [75]. In conclusion, wild seaweed harvesting without an ecological study can pose significant environmental hazards, particularly when performed only for economic profit, as it depletes primary producers, unbalances the food chain, and takes a long time to recover [143]. As a result, by studying natural seaweed species, macroalgal farming can offset the detrimental impact, manage eutrophication problems in ecosystems,

as well as be particularly effective at removing nutrients from water, therefore controlling algal blooms [144,145]. There are nations that have considerable harvesting of seaweed species; however, due to the economic consequences of unregulated harvesting in the past, natural harvesting is mostly conducted in sustainable methods [146]. As a result, seaweed production might be regarded as an ecological service. with economic exploration, if performed correctly [145].

The seaweed ecosystem services value in terms of functions is similar to the freshwater macrophytes and seagrass, and all the marine flora ecosystem appears to be similar between the different types of flora. Seaweed are keystones for the ecosystem maintenance and status quo where they are found. Their economic impact depends on the habitat area, and in the provisioning and regulating services [27,147–149]. In terms of economic values, there are many marine flora impacted by each provider: plankton, coral reefs, seagrass and seaweeds.

## 5. Conclusions

As demonstrated by this review, we observe that seaweeds are a key element to maintain a high number of ecosystems working naturally, supporting the humankind with several ecological services. Thus, the maintenance the status quo in estuarine and coastal areas are needed to protect not only ecosystem but also the human existence. Furthermore, the rehabilitation of estuarine and coastal zones in damaged ecosystems employing various plant native seaweeds provides significant advantages to humans. When ecosystems are subjected to human effects (for example, eutrophication, damming, and species invasions), seaweeds lose their ability to supply ecological services and, in certain cases, can even provide "dis-services" to the humans. Finally, protection of seaweeds biodiversity should be addressed since there is evidence that most of the ecological services that can be outlined are more efficient in areas with richer and more functionally varied ecosystems. In an era where various stresses threaten ecosystems and human life, the conservation of seaweed variety and their usage as suppliers of benefits to people is both a problem and a necessity. Thus, in the future seaweed cultivation must be integrated into damaged zones to recover the ecosystem and make the appearance of invasive species and pathogens difficult.

**Author Contributions:** Conceptualization, J.C.; preparation, J.C. and L.G.; Images, J.C., L.G. and D.P.; Draft Writing, J.C., L.G. and D.P.; writing—review and editing, J.C., L.G., D.P. and L.P. All authors have read and agreed to the published version of the manuscript.

**Funding:** This work is financed by national funds through Foundation for Science and Technology (FCT), I.P., within the scope of the projects UIDB/04292/2020, Marine and Environmental Sciences Centre (MARE), and Associate Laboratory ARNET. João Cotas thanks the European Regional Development Fund through the Interreg Atlantic Area Program, under the project NASPA (EAPA_451/2016). This study had the support of national funds through Fundação para a Ciência e Tecnologia (FCT), under the project LA/P/0069/2020 granted to the Associate Laboratory ARNET.

**Institutional Review Board Statement:** Not applicable.

**Informed Consent Statement:** Not applicable.

**Data Availability Statement:** Not applicable.

**Conflicts of Interest:** The authors declare no conflict of interest.

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
