# Peer review of "Ecosystem Services Provided by Seaweeds"

_2673-9917, doi:10.3390/hydrobiology2010006_

Round 1

Reviewer 1 Report

‘Ecosystem Services Provided by Seaweeds’ by Cotas et al. is a good piece of review. I recommend the publication of the article after a minor revision.

 Abstract should be more appropriate.

Keywords should be selected carefully.

In introduction section, many statements are without proper references.

Some sentences like Line 50-55 are too long, difficult to comprehend.

Introduction completely lacks aims and objectives of the study. A strong justification to the objectives is missing.

Line 62 - Why only web of science and goggle scholar; there are many other sources; mention other sources also.

Line 170-174 is in different font.

Sources of the pictures or figures taken from secondary sources (if any) must be included

Author Response

Reviewer #1:

Comment 1: ‘Ecosystem Services Provided by Seaweeds’ by Cotas et al. is a good piece of review. I recommend the publication of the article after a minor revision.

Answer 1: Thank you for your kind words.

Comment 2:
Abstract should be more appropriate. Keywords should be selected carefully.

Answer 2: We revised the abstract and keywords

Comment 3: In introduction section, many statements are without proper references. Some sentences like Line 50-55 are too long, difficult to comprehend. Introduction completely lacks aims and objectives of the study. A strong justification to the objectives is missing.
Answer 3: We revised the introduction.

Comment 4: Line 62 - Why only web of science and goggle scholar; there are many other sources; mention other sources also.
Answer 4: We added more info, however, the WoS and Google scholar did report the major part of articles, and all the other sources given the same ones of google scholar.

Comment 5: Line 170-174 is in different font.
Answer 5: Thank you for your advise, we format to manuscript template

Comment 6: Sources of the pictures or figures taken from secondary sources (if any) must be included
Answer 6: All the images are originals from the authors

Reviewer 2 Report

Very interesting paper, Ilearned a lot ! very comprehensive and developing the subject widely and precisely.

Some small formatting problems very easy to solve. Examples ;

Put the header of the first table on the same page than data (ll90 and following ones) and same for the drawing

Somme formatting problems in the reference list (capitals, spaces, info missing (ll 645,770, 772, 823, 846, 855, 877-8, 905, ... not comprehensive).

I would have appreciated a small comparison with some other similar ecosystems in order to be able to compare the importance of seaweeds compared, for instance with seegrass meadows ... or coral reefs.

Good job !

Author Response

Reviewer #2:

Comment 1: Very interesting paper, Ilearned a lot ! very comprehensive and developing the subject widely and precisely.

Some small formatting problems very easy to solve. Examples ;

Put the header of the first table on the same page than data (ll90 and following ones) and same for the drawing

Somme formatting problems in the reference list (capitals, spaces, info missing (ll 645,770, 772, 823, 846, 855, 877-8, 905, ... not comprehensive).

I would have appreciated a small comparison with some other similar ecosystems in order to be able to compare the importance of seaweeds compared, for instance with seagrass meadows ... or coral reefs.

Answer 1: Thank you for the reviewer kind words. We did revise the formatting and all references. We added a small paragraph about the ecosystem services of the marine flora.